# Dealing with Plastic Waste from Agriculture Activity

**Teresa Batista** [1,2], **Isabel Pestana da Paixão Cansado** [3,4,*], **Barbara Tita** [5], **Ana Ilhéu** [5], **Luis Metrogos** [2], **Paulo Alexandre Mira Mourão** [4], **João Manuel Valente Nabais** [6], **José Eduardo Castanheiro** [4], **Cátia Borges** [7] **and Gilda Matos** [7]

[1] Instituto Mediterrâneo para a Agricultura, Ambiente e Desenvolvimento (MED), Universidade de Évora, Pólo da Mitra, Apartado 94, 7006-554 Évora, Portugal; mtfb@uevora.pt or tbatista@cimac.pt

[2] Comunidade Intermunicipal do Alentejo Central, Rua 24 de Julho, nº1, 7000-673 Évora, Portugal; luis.metrogos@cimac.pt

[3] LAQV-Requimte, Instituto Mediterrâneo para a Agricultura, Ambiente e Desenvolvimento (MED), Institute for Research and Advanced Training (IIFA), Universidade de Évora, Rua Romão Ramalho, nº59, 7005-671 Évora, Portugal

[4] Instituto Mediterrâneo para a Agricultura, Ambiente e Desenvolvimento (MED), Institute for Research and Advanced Training (IIFA), Universidade de Évora, Rua Romão Ramalho, nº59, 7005-671 Évora, Portugal; pamm@uevora.pt (P.A.M.M.); jefc@uevora.pt (J.E.C.)

[5] Empresa de Desenvolvimento e Infraestruturas de Alqueva (EDIA, SA), Departamento de Ambiente e Ordenamento do Território, Rua Zeca Afonso, nº2, 7800-522 Beja, Portugal; btita@edia.pt (B.T.); ailheu@edia.pt (A.I.)

[6] Comprehensive Health Research Center (CHRC), Departamento de Ciências Médicas e da Saúde, Escola da Saúde e Desenvolvimento Humano, Universidade de Évora, Rua Romão Ramalho, 59, 7000-671 Evora, Portugal; jvn@uevora.pt

[7] Gesamb–Gestão Ambiental e de Resíduos, EIM, Estrada das Alcáçovas, EN 380, 7000-175 Évora, Portugal; c.borges@gesamb.pt (C.B.); gilda@gesamb.pt (G.M.)

**\*** Correspondence: ippc@uevora.pt

**Abstract:** The increase in agricultural production and food quality has forced the growing use of plastics in various activities. The plastic wastes are partially recycled in or outside Portugal; nevertheless, the contaminated wastes are sent to landfill. It is crucial to consider new models for their valorization at a regional level and from a circular economy perspective. In the scope of the Placarvões project, a study was elaborated, which included the types and quantities of plastics used in the irrigation area of the Alqueva Dam, in southern Portugal. The crops that use the most plastic are intensive olive groves, almonds, and table grapes, which represent more than 91% of total plastic waste. The production of activated carbons (ACs) is a solution to avoid plastics landfill. ACs were produced from plastic used on food packaging (PB-Samples) and sheeting film (PS-Samples) by activation with $K_2CO_3$. ACs presented well-developed textural properties (PB-$K_2CO_3$-1:1–700 and PS-$K_2CO_3$-1:1–700 exhibited a volume of 0.32 and 0.25 $cm^3\ g^{-1}$ and an apparent surface area of 723 and 623 $m^2\ g^{-1}$, respectively). Both ACs performed very well concerning four pesticide removals from the liquid phase. This solution is very promising, such these ACs could be applied in effluent treatments on a large scale.

**Keywords:** Placarvões; circular economy; agriculture by-products; Alentejo; Alqueva

## 1. Introduction

The increase in the world's population has led man to improve farming practices to increase agriculture productivity. This was firstly achieved through the exploitation of natural resources, such as water and plant resources, and the excessive use of fertilizers and pesticides. As a way to promote crop protection and increase production, a diversity of plastic containers is used daily in agricultural activities, such as crop containers, bags for pesticides and liquid fertilizers, food bags, cereal bags, greenhouse cover, growing bags, pots, screens, and sticks. The use of different plastics allows to control the increasing

temperature, to reduce the use of water, nutrients, and pesticides, by keeping crops under a small area, such as a greenhouse [1].

The excessive use of plastics in the various stages of cultivation threatens the overall sustainability of our agricultural soils and aquatic environments, due to the plastic waste that persists thereafter the harvest seasons. Around the world, the global use of plastics in agriculture was around 4.4 million tons in 2012 and reached 7.4 million tons in 2019 [1]. Among the most commonly used types of plastics in agriculture are polyethylene (PE), polypropylene, polyvinylchloride, ethylenvinylacetate, polymethylmethacrylate, and polycarbonate [2].

The management of large-sized plastics can produce micro pollution through the accumulation of plastic debris and potentially fragmenting into smaller pieces, classified as particulate plastics, which can be divided into microplastics (fragments < 5 mm) and nanoplastics (fragments < 0.1 μm) [3–6]. A literature revision allows us to state that the majority of microplastics research reported the presence of plastics or microplastics in aquatic environments, especially the marine ones. Wang et al. (2020) presented work concerning the ecotoxicological consequences of microplastics contamination on soil ecosystems, including the effects on soil physical and chemical properties, terrestrial plants, soil fauna, and soil microbes [6]. There is a lack of studies that report soil contamination by microplastics and nanoplastics from agricultural practices.

The plastics degradation takes place very slowly and produces a host of secondary pollutants, which include a diversity of volatile organics, representing a risk of groundwater contamination [7,8]. The presence of microplastics is especially relevant in soils used for human food production, and to better manage and control the plastic residues from agricultural use, the identification of the sources of their inputs into the environment becomes imperative [9].

The amount of agricultural plastic used in the Alqueva Dam irrigation area, one of the biggest irrigated areas in Portugal, in the Alentejo region, in a diversity of applications, is now estimated at 1880 tons per year, and with the growth of the irrigated area, could reach 3500 tons per year over the next few years [10]. The Alqueva Dam irrigation area is shown in Figure 1, the cultures occupying a greater extent of land are presented in Table 1, and examples of dirty waste plastic produced are illustrated in Figure 2.

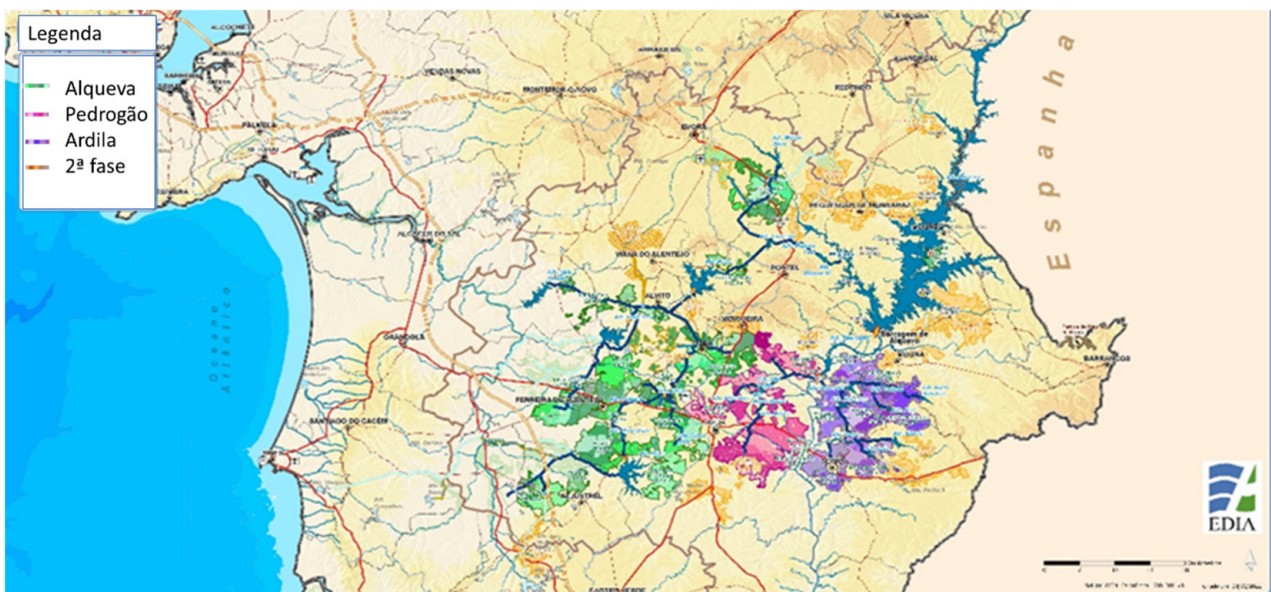

**Figure 1.** Map of Alqueva Dam irrigation area.

**Table 1.** Main cultures practice in the Alqueva Dam irrigation area. In Table 1, ha = 10,000 m$^2$.
* Intensive olive groves.

| Culture | Area (ha) 2020 | Mean Productivity (ton/ha) | Plastic Used (ton/ha/year) | Plastic Wastes (ton/Year) |
|---|---|---|---|---|
| Olives groves * | 68 346 | 10 | 0.03 | 2050 |
| Almond | 15 241 | 3 | 0.03 | 457 |
| Wine grape | 5755 | 10 | 0.003 | 17 |
| Table grape | 414 | 30 | 1.34 | 555 |

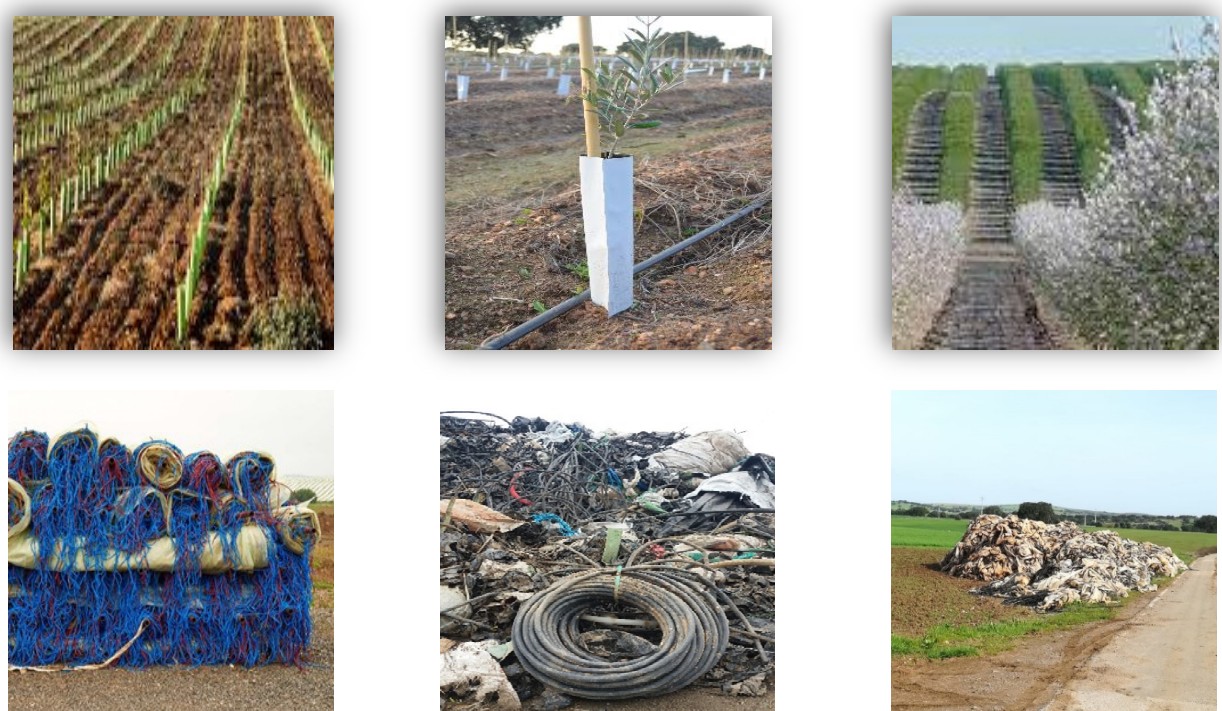

**Figure 2.** Photos from the cultures more present in the Alqueva Dam irrigation region and examples of waste plastics produced.

Agriculture uses a huge amount of conventional PE film on diverse activities. The degradation of PE film can be separated as an abiotic or biotic process. The abiotic is defined as a process caused by environmental factors and the biotic is defined as biodegradation caused by the action of microorganisms present on the soil. However, the degradation process is too long. It could stay in the environment for more than one hundred years, it could be considered inert. Hadad et al., in 2005, presented work concerning the biodegradation of PE. They stated that the PE film could be biodegraded if the right microbial strain was isolated. However, they reported that, after 10 years of a PE sheet incubation in soil, submitted to UV irradiation, less than 0.5% of the PE sheet was degraded. After incubation in moist soil for 12 years, no degradation signals were identified, and only partial degradation was observed in a PE film buried in soil for 32 years [11].

The development of bioplastics emerges as an extraordinary alternative, but their high price and lower reliability of consumers associated prevents its widespread use. A major part of plastics from agricultural activities are not biodegradable and can be treated in a different way, such as reuse for other purposes, recycling, burying plastics on-farm, incineration as a fuel source, or sent to landfill; each of these solutions has its advantages and disadvantages. Agricultural plastics are mostly treated by incineration and mechanical recycling. The increase in the ecological footprint intrinsic due to this treatment and the

presence of dirty agrochemicals in these plastics requires the use of new and more eco-friendly methods [12,13]. The challenge is to ensure that these plastics are reused over a long period and that, at the end of life, they are valued, causing a small impact on the environment. In this perspective, the waste valorization through their transformation into activated carbons (ACs) presents itself as a viable solution.

ACs are material adsorbents that present exceptional physical and chemical properties, such as high surface area, well-developed pore structure (micro-, meso-, and macropores), and a high degree of surface reactivity. ACs are used in a diversity of industrial applications, such as catalysis or support catalyst, energy storage, and as adsorbents for the removal from the liquid or gaseous phases, unwanted odor, pollutants, taste, coloration, heavy metals, organic chemicals such as paint thinners, pesticides, and pharmaceutical and phenolic compounds, among others [13–22].

Several reasons are determinants for the choice of precursor, among which stand out local availability, carbon content, and ease of activation [14–16]. A diversity of low-cost precursors, such as coconut, Tectona Grandis sawdust, pistachio shell, charcoal, cork waste, peat, almond shell, rice husk, bituminous and brown coal, agricultural biowaste and an assortment of synthetic polymers was, already, used on the ACs production [14–17,19–23]. To date, many of these precursors have been used clean, i.e., free of contaminants, such as those present in the precursors used in this work (pesticide residues and herbicides, biomass residues and soil inerts). The innovation of this work is based on the valorization of contaminated agricultural plastics, which, when cannot be directed to a recycling process, are directed to landfills.

With the implementation of new regulations for waste discharge and novel applications, an increase in the global activated carbon market is expected in the coming years. Based on the Fortune Business Insights, the global market for AC was USD 2856.7 million in 2019 and must reach USD 4064.7 million in 2027, which means an increase of 4.8% [24]. However, marketdataforecast.com reports a growth rate of the AC market for the same period of 11.46%, which will reach a value of USD 11.79 billion in 2026 [25].

The continued use of agrochemicals contributes to the contamination of soils and water streams. In this context, a study was carried out, in the Alqueva region, that reported the presence of 25 pesticides. Among the most abundant were bentazone, metolachlor, ter-buthylazine, 2,4-dichlorophenoxyacetic acid (2,4-D), and 4-chloro-2-methyl-phenoxyacetic acid (MCPA). At some sampling points, MCPA showed an individual concentration reaching 580 ng/L [18].

This work uses the concept of circular economy to better manage and control the plastic residues from agricultural use, in particular, the area of influence of Alqueva and it is adjacent irrigated perimeters by the valorization of dirty waste plastics used on agricultural activities through their transformation into ACs, which were subsequently tested for the removal of pesticides from the liquid phase.

## 2. Materials and Methods

### 2.1. Qualitative and Quantitative Identification of Plastics from Agricultural Use

The qualitative and quantitative identification of plastics used in irrigated crops, covering the area of influence of Alqueva and its adjacent irrigated perimeters, were made within the scope of the Placarvões project (from plastic waste to activated carbons). In particular, the plastics used throughout the agriculture cycle, either in the installation or exploitation phases, were identified for each culture in one of the biggest irrigated areas in Portugal, the Alqueva Dam irrigation area. The methodology establishes a reference plot of $700 \times 700$ m (49 ha; 1 ha = 10,000 m$^2$) which is 90% irrigated land and details, for each crop, types of plastic used throughout the culture cycle were identified. The public report, which includes the Life Cycle Assessment of Agricultural Plastics and Regulatory Framework, can be seen elsewhere [10].

### 2.2. Activated Carbon Production

After being used in agricultural procedures, dirty plastics are not in acceptable condition for a future recyclable process. The disposal of these plastics goes through landfill deposition. To promote the reduction of plastics sent to landfill deposition, these wastes were used as a precursor to activated carbons (ACs) production. The box plastics from food packaging and dirty sheeting plastics collected directly from the field were cut into small pieces (around 2.5 cm × 2.5 cm). These pieces were placed into a steel container and mixed with a quantity of the activating agent, i.e., $-K_2CO_3$, in a ratio of 1:1. Subsequently, the steel container was placed into the semi-industrial rotating horizontal tubular furnace, High Temp Technology, TR–334/2018, from Thermolab, used for this purpose. The mixture was submitted to a temperature rate of 10 K min$^{-1}$, until 973 K, under a nitrogen flow and kept for 15 min. The carbonaceous material was cooled down, always under a nitrogen flow rate of 85 cm$^3$ min$^{-1}$, and removed out of the furnace. The excess activating agent remaining on the carbonaceous material was removed by successive washing cycles. ACs obtained from box plastics and sheeting plastics from agricultural use, named PB-$K_2CO_3$-1:1–700 and PS-$K_2CO_3$-1:1–700, were dried, weighed, and stored. The names of the ACs were assigned as follows: PB-$K_2CO_3$-1:1–700 and PS-$K_2CO_3$-1:1–700; PB—plastic box; PS—sheeting plastics; $K_2CO_3$—the activating agent used; 1:1—the ratio between precursor and activating agent and 700–700 °C or 973 K.

The yield obtained with both precursors ranged from 19.2 to 23.1%, which is in the same order of magnitude as the values obtained in previous studies, in which ACs were prepared from clean synthetic polymers, such as from polyethyleneterephthalate, with $K_2CO_3$ or KOH [20].

### 2.3. Activated Carbon Characterization

The textural properties of the ACs were obtained from the analysis of nitrogen adsorption isotherms, carried out at 77 K, on a Quadrasorb gas adsorption manometric equipment from Quantachrome Instruments, USA. The chemical composition was evaluated through elemental analysis (EA-CHNS), using a Eurovector EuroEA 3000, CHNS analyzer, from EuroVector, S.p.A., Italy. SEM was obtained by the Servicio de Apoyo a la Investigación de la Universidad de Extremadura (Badajoz, Spain), using a Quanta 3D FEG-FEI, model HITACHI S-4800, with a Bruker Quantax EDS System. The determination of the pH at the point of zero charge (pHpzc) was obtained based on the mass titration method. Further details concerning the experimental procedure can be consulted elsewhere [17,19].

### 2.4. Activated Carbon Application on Pesticide Removal

The most representative ACs samples prepared, with well-developed porous volume (PB-$K_2CO_3$-1:1–700 and PS-$K_2CO_3$-1:1–700), were tested for 4-chloro-2-methyl-phenoxyacetic acid (MCPA), 2,4-dichlorophenoxyacetic acid (2,4-D), atrazine, and diuron removal from the liquid phase.

The equilibrium time was evaluated by placing suspensions containing 10 mg of ACs on 25 mL of solutions containing the pesticide, at a pH around 7, in a thermostat bath, under agitation, for 48 h [17]. At different time intervals, aliquots were collected, and their absorbance was measured at a wavelength of 228 and 279 nm for MCPA; 230 and 284 nm for 2,4-D; 226 and 264 nm for atrazine; and 212 and 249 nm for diuron, using a Nicolet Evolution 300 UV–Vis spectrophotometer. The maximum amount of each pesticide adsorbed was evaluated using two procedures. First, for MCPA and 2,4-D, a fixed amount of AC (around 10 mg, weighed rigorously) was added to 25 mL of an aqueous solution containing 1.25 mmol L$^{-1}$ for MCPA and 1.13 mmol L$^{-1}$ for 2,4-D, that is 250 mg L$^{-1}$. Second, solutions were prepared with a concentration of 0.31 mmol L$^{-1}$ for MCPA, 0.28 mmol L$^{-1}$ for 2,4-D, that is 62.5 mg L$^{-1}$, and 0.14 mmol L$^{-1}$ for atrazine and 0.12 mmol L$^{-1}$ for diuron, that is 30 mg L$^{-1}$. In this case, a fixed amount of AC (around 10 mg) was added to 200 mL of the aqueous solution containing the pesticide.

All suspensions were placed in a thermostatic bath for 24 h, at 298 K. The residual pesticide amount was evaluated, in triplicate, using an external pattern and all calibration lines used allowed the achievement of excellent correlation coefficients ($R_2 > 0.999$). The pesticides amount adsorbed was calculated based on Equation (1).

$$Q_{ads} = V \frac{(C_0 - C_{eq})}{m} \tag{1}$$

where $Q_{ads}$ is the uptake of adsorbate at equilibrium expressed by mass of AC, *m* is the mass of AC, $C_0$ is the initial concentration, and $C_{eq}$ is the equilibrium concentration of pesticide [17].

### 3. Results

*3.1. Plastics Used in Different Cultures*

Currently, the agricultural plastic volume in the Alqueva irrigation area is 1.880 tons per year, and with the expected growth of the irrigated area, it can reach 3.500 tons per year. The plant protectors are the plastic with the greatest influence, 56% in the total, followed by the plastic film. These two types of plastic represent about 85%. The crops that use the most plastic are intensive olive groves, almonds (plant protectors), and table grapes (plastic film), and represent more than 91% of the total plastic wastes of the agricultural area. The plastics amount used throughout the agriculture cycle, either in the installation or exploitation phases, was previously identified for each culture [10]. The full data is freely available for those interested in knowing more about this particular topic. In a nutshell, Figure 3a,b show the total amount of plastic, expressed in percentage, by culture type and the typology of plastic used by culture type, respectively.

The Alentejo cannot afford not to value some wastes, thus transforming them into valuable materials that allow increasing regional income. Data obtained in this study was the starting point of this work, which mobilized the scientific community and entrepreneurs from the region, for the dirty waste plastic valorization, promoting the application of local and regional circular economy.

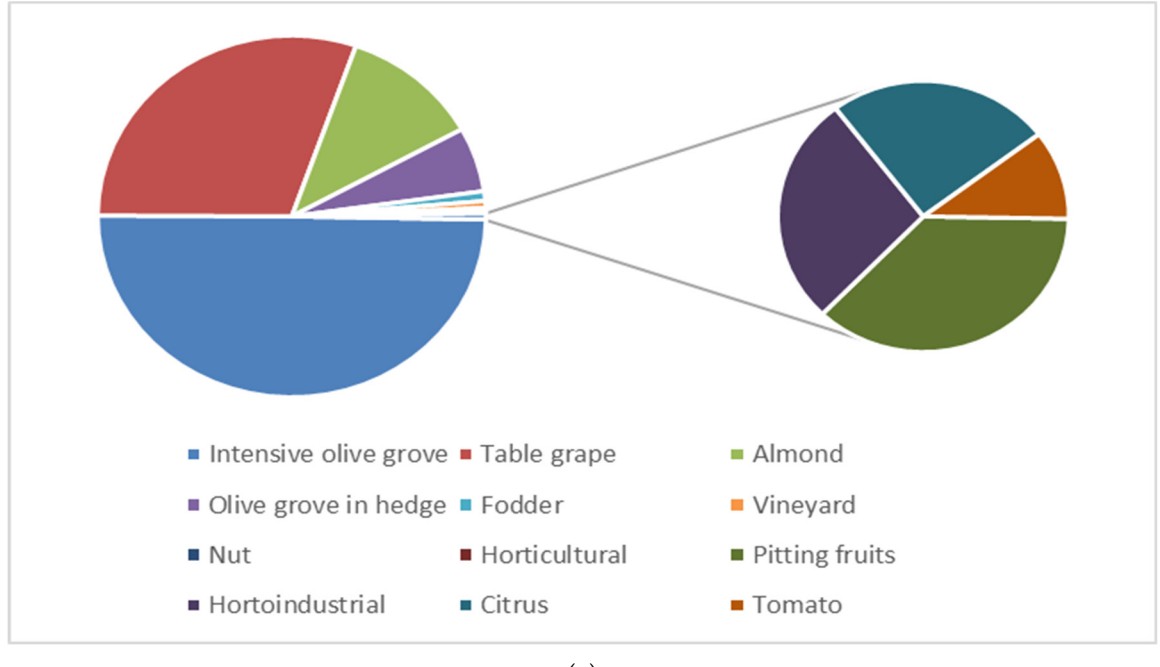

(**a**)

**Figure 3.** *Cont.*

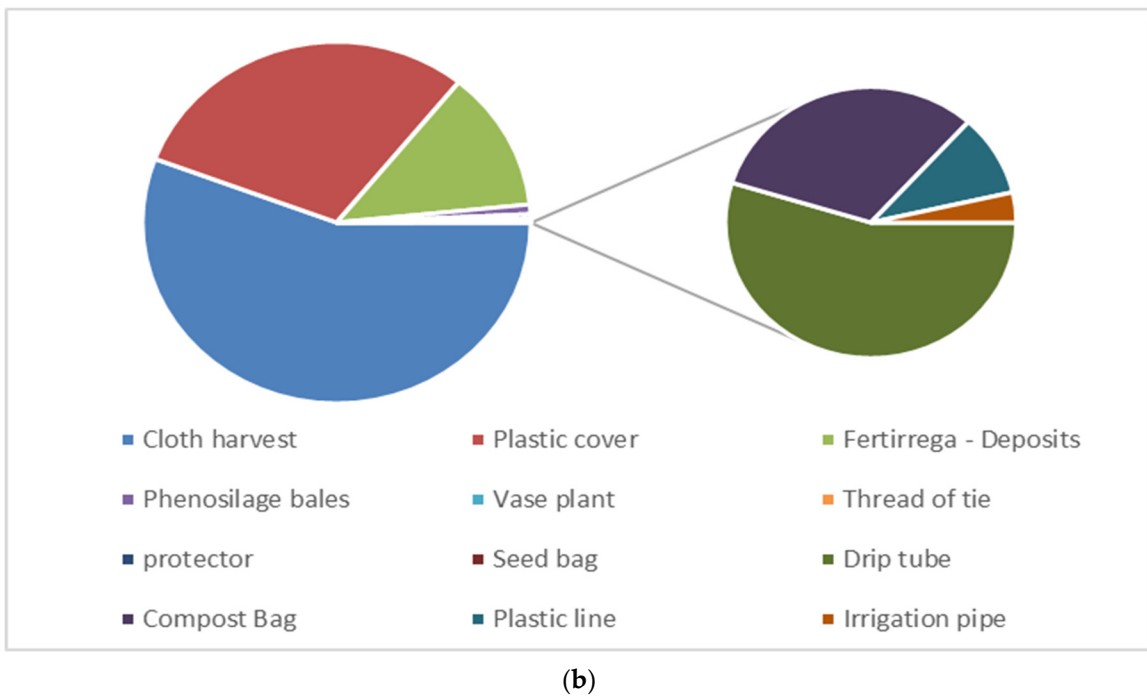

**(b)**

**Figure 3.** (**a**) Percentage amounts of the plastic used by culture type in the Alqueva irrigation area. (**b**) Percentage amount of each kind of plastic used by culture type in the Alqueva irrigation area.

### 3.2. Activated Carbon Production and Characterization

A diversity of ACs was produced from box plastics used on the food packaging and sheeting plastics used to cover crops, based on a preliminary study that allowed the optimization of several parameters of the production method that influence the final characteristics of ACs [23]. The ACs produced by chemical activation with $K_2CO_3$ presented a well-developed porous structure. PB-$K_2CO_3$-1:1–700 and PS-$K_2CO_3$-1:1–700 exhibited a microporous volume of 0.32 and 0.25 $cm^3\ g^{-1}$, and an apparent surface area of 723 and 623 $m^2\ g^{-1}$, respectively, as presented in Table 2.

**Table 2.** Textural and chemical characteristics of the PB-$K_2CO_3$-1:1–700 and PS-$K_2CO_3$-1:1–700. In Table 2, the $A_{BET}$ ($N_2$) is the apparent surface area, Vs and $V_0$ are the total and microporous volume, respectively, and $L_0$ is the mean pore size.

| Activated Carbon | $A_{BET}/$ $m^2\ g^{-1}$ | $V_s/$ $cm^3\ g^{-1}$ | $V_0/$ $cm^3\ g^{-1}$ | $L_0/$ nm | C/ wt% | $O_2/$ wt% |
|---|---|---|---|---|---|---|
| PB-$K_2CO_3$-1:1–700 | 723 | 0.32 | 0.18 | 1.22 | 75.1 | 23.0 |
| PS-$K_2CO_3$-1:1–700 | 623 | 0.25 | 0.12 | 3.11 | 71.1 | 25.8 |

Both ACs were characterized by SEM, and some illustrative images are presented in Figure 4a,b. It can be seen that both ACs have similar structures, despite that both ACs were prepared from different precursors which were contaminated with soil debris. ACs presented a nitrogen content higher than 70%, and the oxygen content determined by elemental analysis was relatively high (PB-$K_2CO_3$-1:1–700–23.0% and PS-$K_2CO_3$-1:1–700–25.8%), which influences its functional surface chemistry. The ACs presented a basic character, confirmed by the determination of the point of zero charge, that is, pHpzc $\cong$ 9.

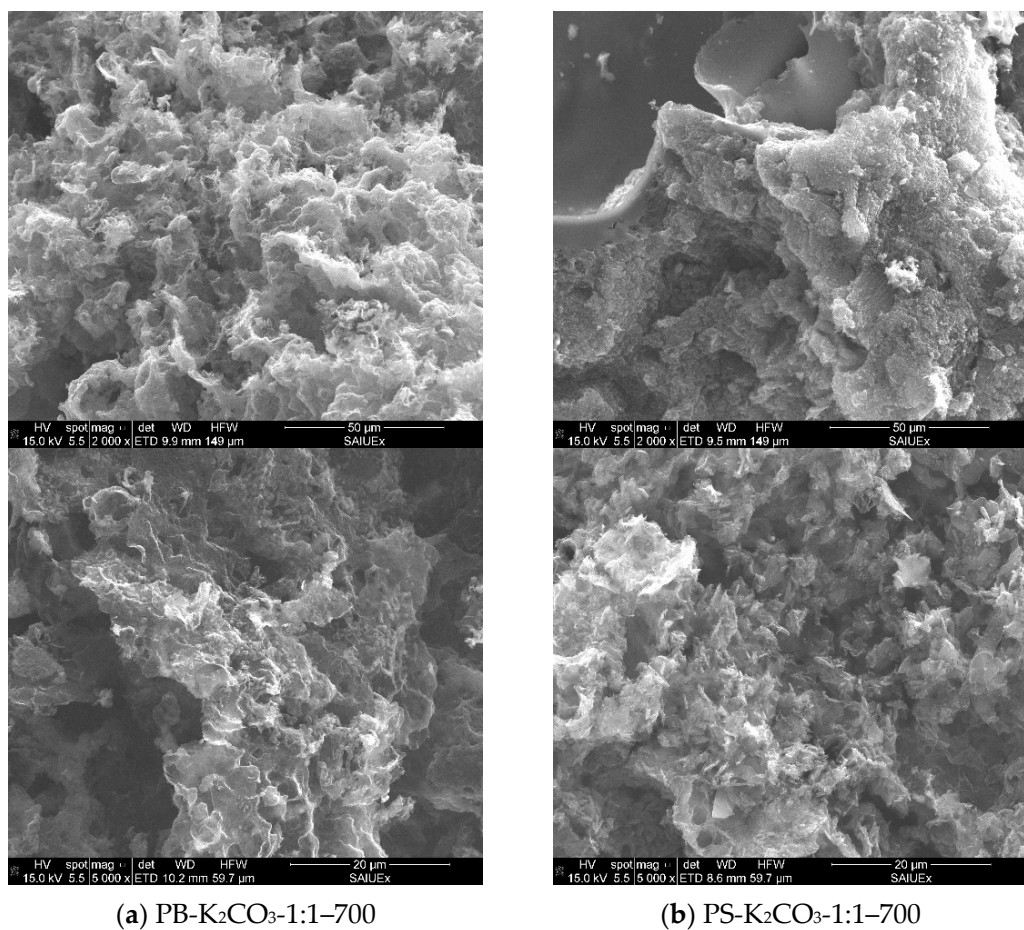

(**a**) PB-K$_2$CO$_3$-1:1–700          (**b**) PS-K$_2$CO$_3$-1:1–700

**Figure 4.** SEM image obtained on the ACs prepared from packing box plastics (**a**) and plastic sheeting (**b**), by chemical activation with K$_2$CO$_3$, at 973 K. On the images, each mark corresponds to 50 μm and 20 μm, respectively, on the first and second lines.

*3.3. Pesticides Removals from the Liquid Phase, at 298 K, Using Activated Carbons*

The slightly basic surface character of both ACs anticipates their good performance in the adsorption of different acidic pesticides. PB-K$_2$CO$_3$-1:1–700 and PS-K$_2$CO$_3$-1:1–700 were tested on the MCPA, 2,4-D, atrazine, and diuron removals from the aqueous phase. The equilibrium of the removal process was achieved after a contact time of less than 24 h and using concentrated solution; this was 1.25 mmol L$^{-1}$ for MCPA and 1.13 mmol L$^{-1}$ for 2,4-D, corresponding to a mass of 250 mg L$^{-1}$. It can be highlighted that after a contact time of 120 min, 60% of the total amount of MCPA and 2,4-D was already adsorbed, as presented in Figure 5.

It must be highlighted that in a situation of accidental spills, where pesticides are found in soils or aquifers at high concentrations, their removal or immediate containment becomes predominant. In these situations, the use of ACs in filter systems or by direct mixing with soil is an effective way to contain or remove pesticides, avoiding their dispersion in the surrounding areas.

Figure 6a,b present the maximum amount of four pesticides adsorbed on PB-K$_2$CO$_3$-1:1–700 and PS-K$_2$CO$_3$-1:1–700. The data presented in Figure 6a, concerning the 2,4-D and MCPA adsorption, was obtained from concentrated solutions. Figure 6b presents also the maximum amounts of pesticides obtained on both ACs, but the data presented, concerning the MCPA and 2,4-D adsorption, were obtained from diluted solutions. However, the amount of 2,4-D and MCPA in the solution in contact with the ACs was similar to that used in tests carried out from concentrated solutions.

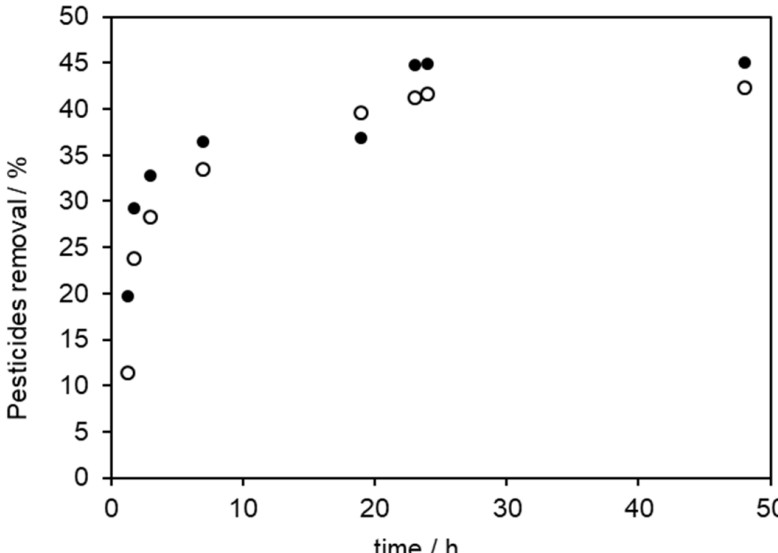

**Figure 5.** Adsorption kinetics of 2,4-D and MCPA, obtained on PB-K$_2$CO$_3$, at 298 K. Open data points represent the adsorption of 2,4-D and filled data points the adsorption of MCPA.

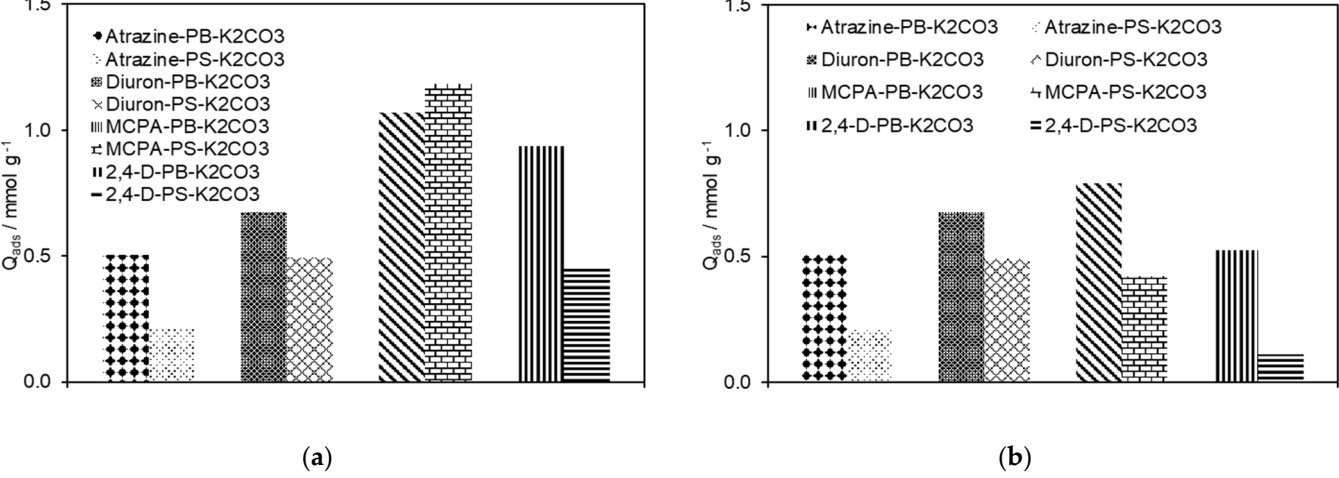

(**a**)                                             (**b**)

**Figure 6.** Maximum adsorption capacities obtained for the four pesticides on PB-K$_2$CO$_3$ and PS-K$_2$CO$_3$. (**a**) Data concerning MCPA and 2,4-D were obtained from concentrated solutions. (**b**) Data concerning MCPA and 2,4-D were obtained from diluted solutions.

Previous works have shown that the removal of MCPA and 2,4-D by ACs with a predominantly basic nature presents better performance if performed from an acid medium (pH $\leq$ 5) [14]. It was reported that basic solutions improve atrazine adsorption, pH of the solution has no significant influence on diuron adsorption [19], and acidic conditions favored MCPA and 2,4-D adsorption, on ACs presenting a slight basic character [20]. However, these tests were carried out from neutral solutions, to simulate and optimize conditions that promote the removal of these pesticides when simultaneously present in the same solution.

As presented in Figure 6a, both ACs showed a good performance concerning the MCPA and 2,4-D removal from the liquid phase. The maximum adsorption capacity increased from atrazine $\leq$ diuron $\leq$ 2,4-D $\leq$ MCPA. The data concerning the 2,4-D and MCPA were obtained from concentrated solutions (1.25 mmol L$^{-1}$ for MCPA and 1.13 mmol L$^{-1}$ for 2,4-D). In this situation, adsorption was controlled by the ACs porous structure and accessibility of pesticides to the inside pores. PB-K$_2$CO$_3$ exhibited a higher adsorption capacity concerning atrazine, 2,4-D, and diuron, when compared with PS-K$_2$CO$_3$. The exception was the MCPA, for which maximum adsorption removal was almost similar on

both ACs. These were expected results, as PB-$K_2CO_3$ presented a higher apparent surface area and pore volume ($A_{BET}$–723 $m^2$ $g^{-1}$; V–0.32 $cm^3$ $g^{-1}$) when compared with PS-$K_2CO_3$ ($A_{BET}$–623 $m^2$ $g^{-1}$; V–0.25 $cm^3$ $g^{-1}$), and both characteristics favored the MCPA and 2,4-D adsorption.

Atrazine and diuron presented lower water solubility (30 and 42 mg $L^{-1}$) than MCPA and 2,4-D (825 and 677 mg $L^{-1}$, respectively). Dilute solutions of MCPA and 2,4-D were used to obtain data in similar conditions to those used with atrazine and diuron; see data presented in Figure 6b. A total of 10 mg of ACs were placed in equilibrium with 200 mL of each solution of pesticides, allowing a ratio of $6.2 \times 10^{-3}$ mmol of MCPA, $5.6 \times 10^{-3}$ mmol of 2,4-D, $2.8 \times 10^{-3}$ mmol of atrazine, and $2.6 \times 10^{-3}$ mmol of diuron per gram of AC used. Figure 6b confirms that PB-$K_2CO_3$ shows the maximum adsorption capacity concerning the four pesticides. From the diluted solutions, the maximum amount of MCPA and 2,4-D removed by both ACs was far from the amount achieved from concentrated solutions, as shown in Figure 6a.

The comparison between Figure 6a,b makes it clear that pesticides diffusion on the diluted solutions was the main factor controlling the kinetics of the process and the ACs maximum adsorption capacity.

## 4. Discussion

The report prepared during the Placarvões project made it very clear that predominant crops in the Alqueva region, in Alentejo, are intensive olive grove, table grape, and almond. Following this increase, the use of plastics in these cultures will also suffer a significant increase. The most used types of plastic in these crops are cloth harvest, plastic cover, and fertirrega deposits. The identification of the quantities and types of plastics used in the different cultures will allow the delineation of strategies for the disposal of their wastes. A way to treat these dirty plastics was evaluated through their transformation into ACs, bearing in mind that contaminated plastics will hardly meet the requirements for recycling processes.

A large amount of low-cost ACs was produced from dirty waste plastics, collected directly from the field, or from plastic used in food packaging, by chemical activation with $K_2CO_3$, at 973 K. The ACs produced were successfully used for the removal of MCPA, 2,4-D, atrazine, and diuron from the liquid phase. These presented a high maximum adsorption capacity concerning MCPA and 2,4-D removal from concentrated solutions, which may correspond to situations of pesticide spillage.

In the presence of high amounts of pesticides on soil, the action time should be short to avoid infiltration and dispersion in surrounding areas. In situations of spills of diluted solutions, or, when it is necessary, a continuous pesticide removal from effluents, the ACs produced also presented a very good performance. In these situations, the use of ACs filters is an effective way of removing pesticides. ACs have the particularity of being able to be regenerated and reused several times with considerable performances. From this point of view, the whole proposal to reduce plastic wastes, through their transformation into ACs, for application in pesticides removal from the liquid phase, which can be regenerated and reused, follows the principles of the circular economy.

## 5. Conclusions

The major conclusion of the work now reported is that dirty plastics can be valorized through their transformation into activated carbons (ACs) adsorbents, which performed well on MCPA, 2,4-D, atrazine and diuron removal, from the liquid effluents. The results obtained show that these ACs, which presented high apparent surface area and porous volume, can be used either in emergency spills situations or as filters for the continuous removal of pesticides from large volumes of contaminated water.

**Author Contributions:** Conceptualization, B.T., T.B., C.B., I.P.d.P.C.; methodology, A.I., P.A.M.M., B.T., G.M.; software, J.E.C., P.A.M.M.; validation, J.M.V.N., P.A.M.M., I.P.d.P.C.; formal analysis, I.P.d.P.C., P.A.M.M., J.M.V.N.; investigation, P.A.M.M., I.P.d.P.C., J.E.C., J.M.V.N.; writing—original

draft preparation, G.M., I.P.d.P.C.; writing—review and editing, I.P.d.P.C., T.B., J.M.V.N., P.A.M.M.; project administration, T.B., B.T., A.I., L.M., I.P.d.P.C.; funding acquisition, T.B., B.T., C.B., I.P.d.P.C., L.M., P.A.M.M. All authors have read and agreed to the published version of the manuscript.

**Funding:** This research was funded by Fundo Ambiental (Environmental Fund of Portugal) with National (OE) funds, 2018. Project–"Placarvões: De plásticos a carvões ativados".

**Institutional Review Board Statement:** Not applicable.

**Informed Consent Statement:** Not applicable.

**Conflicts of Interest:** The authors declare no conflict of interest.

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
