# Peer review of "Dealing with Plastic Waste from Agriculture Activity"

_agronomy, doi:10.3390/agronomy12010134_

Round 1
Reviewer 1 Report
Comments to the Author:
The article entitled “Dealing with plastic waste from agriculture activity” by Tita and coauthors provided a method to deal with the plastic wastes from agriculture activity. The activated carbons were developed with high surface area from plastic wastes, and then used for waste water treatment to remove like the pesticides. Overall, I recommend publication of this work in journal Agronomy after following revisions.
Comments:
Line 30, seems “it was elaborate…” has grammatical error. Please check.
Table 1, the three-line table is suggested.
Line 105-Line 110, can authors add some brief introduction of activated carbons? Apart from use as absorbents, they also can serve as like catalysts, etc. Additionally, are there any reported works involving the production of activated carbons from waste plastics? Any research gap? If can add these, it may strength the novelty of this work.
Line 150, please provide the full name of PB/PS.
Line 153, use subscript of 3 in K2CO3.
Line 157, what was the flow rate of nitrogen gas?
Line 160, how about the yield of activated carbons?
Line 159-160, the 700 should denotes 700 ºC, please label.
Line 248, did the Pesticides removals run at room temperature? Please add this.
Did the K2CO3 can react well with the plastic feed during the preparation of ACs? As if some K2CO3 left after pyrolysis, it may go into the waste water during the Pesticides removals runs.
Figure 6, the fill patterns look confusing, can authors re-select the patterns to better tell the entries?
Reference section: please uniform the format of references, for example, some DOI was attached to the reference, but some are not.
Author Response
Dear Reviewer,
The authors appreciated and thank all suggestions and comments from the reviewer.
All comments and suggestions have been completely answered and discussed throughout the document and are identified.
As reviewers pointed out that English writing should be improved, the authors asked for help in improving writing for a person of English origin. The changes made all along the text are identified.
For other details, please see the attachement.
Isabel Cansado

Reviewer 2 Report
A pretty nice work in plastic upcycling and the application of the final products-Activated carbon. I suggest to publish this work if the following issues could be addressed.1
- first of all, the English of the manuscript could be further improved
- The experimental section could be more detailed.
- It would be great to have full characterization of the AC derived from plastic such as surface area, pore, functional groups.
Author Response
Dear Reviewer,
The authors appreciated and thank all suggestions and comments from the reviewer.
All comments and suggestions have been completely answered and discussed throughout the document and are identified.
As reviewers pointed out that English writing should be improved, the authors asked for help in improving writing for a person of English origin. The changes made all along the text are identified.
For other details, please see the attachment.
Isabel Cansado

Round 2
Reviewer 2 Report
my comments has been well responsed